# High-Temperature Monitoring in Central Receiver Concentrating Solar Power Plants with Femtosecond-Laser Inscribed FBG

**DOI:** 10.3390/s21113762

**Published:** 2021-05-28

**Authors:** Roberto Rodríguez-Garrido, Alejandro Carballar, Jonathan Vera, José González-Aguilar, Adeodato Altamirano, Antonio Loureiro, Daniel Pereira

**Affiliations:** 1Electronic Engineering Department, Universidad de Sevilla (E.T.S. de Ingeniería), C/Camino de los Descubrimientos, s/n, 41092 Sevilla, Spain; rbt151292@gmail.com; 2National Renewable Energy Center (CENER), Solar Thermal & Thermal Energy Storage Department, C/Isaac Newton, 4. Pabellón de Italia, Planta 5 SO, 41092 Sevilla, Spain; jvera@cener.com; 3High Temperature Processes Unit, IMDEA Energy Institute, Avda. Ramón de la Sagra, 3, 28935 Móstoles, Spain; jose.gonzalez@imdea.org; 4MESUREX, C/Marie Curie 4, D10 (Parque Técnológico de Andalucía), 29590 Málaga, Spain; dato.altamirano@mesurex.com; 5EGATEL, C/Avda. Ourense 1 (Parque Tecnolóxico de Galicia), 32901 San Cibrao das Viñas, Spain; loure@egatel.es; 6COBRA Instalaciones y Servicios, C/Cardenal Marcelo Spínola 10, 28016 Madrid, Spain; dpereira@grupocobra.com

**Keywords:** high-temperature sensing, optical fiber sensor, femtosecond fiber Bragg grating, harsh environment, concentrating solar power

## Abstract

This work deals with the application of femtosecond-laser-inscribed fiber Bragg gratings (FsFBGs) for monitoring the internal high-temperature surface distribution (HTSD) in solar receivers of concentrating solar power (CSP) plants. The fiber-optic sensor system is composed of 12 FsFBGs measuring points distributed on an area of 0.4 m^2^, which leads to obtain the temperature map at the receiver by means of two-dimensional interpolation. An analysis of the FsFBG performance in harsh environment was also conducted. It describes the influence of calibration functions in high-temperature measurements, determines a required 10 nm spectral interval for measuring temperatures in the range from 0 to 700 °C, and reveals wavelength peak tolerances in the FsFBG fabrication process. Results demonstrate the viability and reliability of this measuring technique, with temperature measurements up to 566 °C.

## 1. Introduction

Concentrating solar power (CSP), also known as solar thermal electricity (STE), is one of the renewable-energy technologies capable of large-scale and dispatchable power production. Amongst them, Central Receiver System (CRS), or Solar Tower, is considered to have the highest solar-to-electricity conversion efficiency, since it allows us to achieve high temperatures. In CRS–CSP plants, moving mirrors or heliostats concentrate direct solar radiation on a solar receiver located on top of the tower. The concentrated solar energy is transferred as thermal energy to a liquid medium or heat-transfer fluid. The hot liquid is used for producing steam and then electricity by means of a Rankine cycle, or simply stored for future electricity generation according to the demand [1,2,3]. More than 1300 MWe of CRS–CSP plants has been installed all around the world, including countries such as Spain, Israel, United States, China, South Africa, Morocco, or Chile [4]. Figure 1 shows the Crescent Dunes Solar Power Plant under operating conditions and under maintenance stop.

Nitrate-based molten salts (MS) are simultaneously used as heat-transfer fluid and thermal-storage medium in commercial CRS–CSP plants. Here MS are heated in the solar receiver, flowing inside pipes of the thermal panels, and then diverted toward heat exchangers. Mean outlet temperature in the solar receiver is typically about 550 °C. Operation with MS are constraint by their metal corrosion capability, which requires special selection of materials in contact with the MS, and working temperature limits [5,6]. MS must be kept in liquid state at temperatures higher than their melting temperature range (260–270 °C) usually working over 290 °C, otherwise undesirable pipe blocking could be occurred. Besides, heating MS (nitrates) to elevated temperatures (above 575 °C) promotes their thermal decomposition. All these features must be taken into account in the daily start-up and operation of CSP plants. In particular, it demands to get a uniform high-temperature surface distribution (HTSD) around 565 °C on the exposed area of the solar receiver, avoiding hot spots. This is obtained by an active control of heliostats to provide an adequate concentrating solar flux distribution and a real time HTSD monitoring, being both aspects crucial for efficiency, cost, and maintenance of CSP plants. [7].

High-temperature monitoring under high fluxes such as the ordinary working conditions in CRS–CSP plants is extremely challenging due to the harsh environment in terms of temperature and high solar radiation fluxes. Current temperature measurement systems are classified in direct, indirect and combined, and of which selection depends on the receiver type and the measurement purpose. Direct measure systems are composed of one or several sensors distributed in the aperture plane or surface receiver. The sensor directly delivers a signal that is a function of the temperature. Thus, temperature maps are created by spatial interpolation of the measured points. Indirect measure systems determine the temperature through the correction of the flux emitted by the receiver surface thermography [8]. Although infrared cameras are easy to integrate in the System of Control and Data Acquisition (SCADA) of the whole plant, they are expensive, and depending on receiver configuration, the temperature measurement could lead to a high level of uncertainty. Combined systems, which are usually used in commercial receivers, are those that use direct and indirect systems simultaneously.

Among direct measure systems, K-type thermocouple is the most utilized sensor for this application [9], which is able to measure temperature approximately up to 1372 °C. However, it has a limited lifetime, slow time response, troublesome assembly, and they occasionally provide wrong measurements and false alarms. These disadvantages make thermocouple cost prohibitive if they are deployed in a large number. On the other hand, the development of reliable fiber-based temperature sensors capable of operating up to 1000 °C is currently an available technology [10]. Fiber-based high-temperature sensors are applied in a wide range of fields, such as combustion (in combustors and gas turbines), nuclear reactors, and chemical reactors [11,12,13,14,15,16,17].

Fiber-optic sensors have intrinsic advantages, such as passive nature, lightweight, low loss, small size, fast sensor time response, non-affected by external electromagnetic fields [18,19], and capability to operate in harsh environments [10]. Additionally, fiber Bragg gratings (FBG) are wavelength encoded (i.e., the wavelength of reflected light depends on the grating period of the fiber core refractive index modulation), allowing for multiplexing capability and thus multipoint-temperature measurements. Conventional type I photosensitive FBGs are not suitable to operate at high temperature because they strongly degrade at temperatures higher than 200 °C [20,21]. To overcome this issue, type II FBG technology offers two main options. The first one consists of Bragg gratings inscribed in an optical fiber with a femtosecond laser (FsFBG—femtosecond-laser inscribed FBG) [22,23,24,25]; and the second lies in Bragg gratings regenerated by means of a thermal process (RFBG—regenerated FBG) [26,27,28,29]. While FsFBGs exhibit higher refractive index modulation, RFBGs retain precise phase information along the grating length [30]. It has been demonstrated that FsFBGs are able to operate up to 1000 °C, whilst RFBG can withstand temperatures as high as 1295 °C [26,27,28,29].

This work reports the use of a fiber-optic sensor system, based on FsFBGs, to monitor the HTSD of a central receiver prototype installed in a concentrating solar tower facility. For the measurements, two FsFBG arrays (with six-point sensors each) have been installed in the central receiver prototype and HTSD maps have been determined for different irradiance distributions on the solar receiver. Results demonstrate the reliability of the temperature measurement concept based on fiber-optic sensors in CSP, although there are several critical aspects of the design for improvement.

## 2. Experimental Setup

This section presents the design of the fiber-optic sensor system, the description of the instrumentation in the solar receiver prototype, and their installation and commissioning at the tower of a CSP facility to monitor the HTSD when heliostats concentrate solar radiation on it.

### 2.1. Fiber-Optic Sensor System Design

The fiber-optic sensor system consists of 12 FsFBGs and a FBG optical interrogator (at which the FsFBGs are connected). The optical interrogator collects and measures the reflected spectral data from the FsFBGs and transforms the wavelength shifts into temperatures. The FsFBG-based sensor experimental setup is illustrated in Figure 2. The optical design and wavelength planning of the FsFBGs are constrained by the spectral range of operation of the optical interrogator. In this work, a commercial optical interrogator from HBM (model FS22) is used to record the reflection spectrum from FsFBGs. This device has a spectral range of 100 nm (between 1500 and 1600 nm, as it is shown in Figure 2), a resolution of 1 pm, and a sampling rate of 1 sample per second. Due to the limited spectral range of the optical interrogator, and to prevent spectral overlapping, the number of FsFBGs in the fiber-optic sensor system was optimized to 12.

The manufacturing of two commercial probes from FemtoFiberTec with 6-FsFBGs sensors each was considered (instead of only one probe with 12-FsFBGs sensors), since this choice will ease the future instrumentation process of the central receiver prototype and will make the sensor configuration more robust against possible breaks and failures. In the two probes, the FsFBGs were inscribed in the single mode fiber using the femtosecond point by point FBG manufacturing process using a laser beam from an 800 nm Titanium Sapphire laser in order to focus tightly into the fiber core [31,32]. FBGs inscribed by infrared femtosecond lasers are thermally stable at temperatures up to 900 °C [33]. FsFBGs are designed for temperature measurements up to 700 °C and short-term operation up to 1000 °C. First probe incorporates FsFBG#1–#6 with wavelength peaks located from 1501 to 1538.5 nm, with 7.5 nm spacing, and second probe incorporates FsFBG#7–#12 with wavelength peaks located from 1546 to 1583.5 nm, with 7.5 nm spacing. Each probe is packaged with 3 m high temperature stainless steel tube (stable up to 1000 °C) and has 5 m corrugated pigtail with FC/APC connection. Table 1 shows the wavelength specifications for the set of the 12 FsFBGs. 

Manufacturer provided the two probes with 6-FsFBGs sensors each, joined the certificate of calibration of each supplied FsFBG#i. This document specifies the peak wavelength as a function of temperature, taking as a reference value the wavelength peak at 22 °C, λ_ref_. Table 1 lists the reference wavelengths and temperature sensitivity for the FsFBG#i used in this work. From several controlled measurement points from 300 to 600 °C, in 50 °C steps, wavelength–temperature calibration functions, obtained by 3rd-order polynomial function fitting, were determined and used to calculate local temperature from the corresponding wavelength shifts. The corresponding calibration curves for each FsFBG#i sensor are depicted in Figure 3.

### 2.2. Central Receiver Prototype Instrumentation

The central receiver prototype was designed to operate in conditions closed to those found in industrial solar receivers and provides similar thermal features. It was mainly composed of two squared INCOLOY 625 plates 6 mm in thickness and 1300 mm in side. The front and exposed surface was coated by PYROMARK 2500. These material and high-temperature paint are commonly used in commercial receivers. The INCOLOY 625 alloy is used in CRS–CSP plants because it is a highly durable stainless steel alloy. The main reason for using stainless steel alloys for high temperatures is to prevent corrosion and oxidization. PYROMARK 2500 is a high-quality black primer, which increases the optical absorptivity of the surface in which it is applied. Receiver efficiency is directly proportional to absorptivity in the receiver surface, so high absorptivity is desired for the receiver surface. Moreover, the PYROMARK 2500 primer also presents a relatively low emissivity, which reduces the heat loss due to radiative losses compared to other primers and/or material surfaces.

Due to the high amount of power concentrated into the solar receiver, the absence of heat-transfer fluid or refrigeration may cause ultra-high temperatures on the receiver walls within seconds. To withstand the extreme conditions in terms of requested irradiance and temperature, an air-cooling system was specifically designed. Thus, the receiver prototype consists of a front plate that will be exposed to the concentrated solar radiation, while the rear plate will close the prototype in order to enable the forced cooling by air (the two plates are separated 20 mm). A single-inlet fan (36.5 kVA, electric power) located at the solar tower ground propels air though a 60 cm stainless-steel pipe toward the prototype bottom. The cold air flowed through the gap between the prototype plates until the exit located at prototype top. The fan was able to provide a maximum air flowrate of 1.6 m³/s and a pressure drop of 1.5 kPa. The temperature difference between front and rear plates (rear plate is not heated by concentrated solar rays) entails a limit in the concentrated flux allowed in the front plate, which is widely accepted to be around 1200–1300 kW/m^2^. These limitations were considered as a design basis for the receiver prototype and the sensors used, since these conditions (580 °C and 1200 kW/m^2^ of solar flux concentrated in the front plate) will never be overcome in commercial CSP plants. Hence, the cooling system will emulate the circulation of heat-transfer fluid in commercial CSP plants and will keep the whole system and sensors in the desired operation regime, providing a flexible and robust solution and avoiding a complex and expensive prototype based on molten salts (since they imposed serious constraints due to its high melting point and potential environmental and safety issues).

The receiver prototype was instrumented with the FsFBG-based fiber-optic sensor system. The 12 FsFBGs were distributed along the surface of the central receiver prototype in order to obtain 12 local temperature measurements. Figure 4a shows the layout of the fiber-optic sensor system on the front plate of the receiver prototype. The locations of each FsFBG#i are depicted in Figure 4a with black diamonds, whilst the drawing of the two fiber-optic probes with 6 FsFBGs each on the internal surface of the front plate is shown with blue lines. Moreover, the prototype incorporates 24 thermocouples distributed on the internal surface of the front plate of the receiver prototype, as shown in Figure 4a with red circles. Figure 4b shows a CAD illustration of the whole prototype. 

FsFBG sensors are attached in the internal surface of the front plate by mechanical pieces specially designed to keep a plate–sensor direct contact. The two FsFBG probes are placed on the plate in a particular arrangement to avoid an excessive curvature of the cable, as shown in Figure 5a. Figure 5b shows the back of the rear plate with the instrumentation corresponding to the 24 thermocouples, and Figure 5c shows the final arrangement of the central receiver prototype in its definitively location for experiments where the forced air-cooling system can be observed.

### 2.3. Installation in the CSP Plant

The central receiver prototype was installed in the Very High Concentration Solar Tower (VHCST) facility at IMDEA Energy, Móstoles, Spain. This facility has a solar field composed by 169 mono-facet heliostats of 3 m^2^ each (1.6 m × 1.9 m) and an 18 m height tower with 2 testing platforms [34,35]. The central receiver prototype was placed in the testing platform located at 12.1 m. In order to guarantee the durability of the equipment and to avoid possible accidents due to its exposure to high solar fluxes, a thermal shielding mainly composed of high-temperature ceramic fiber boards was mounted at the front of the testing platform. Figure 6 shows the details for the installation of the receiver prototype at the tower, the relative position of heliostats with respect to the solar receiver, and a detail of the thermal shielding. At this point, it is important noting that the central receiver prototype has a 26.6° tilt with respect to the vertical in order to maximize the normal solar radiation incidence from heliostats.

## 3. Results

The experimental tests were conceived to reproduce the working conditions reached in commercial solar receivers in terms of irradiance and temperature. Three trials were carried out in order to analyze the operation and performance of the FsFBG-based sensor system.

The first trial aimed at comparing temperatures measured by FsFBGs and thermocouples to verify the correctness and goodness of the results. Figure 7 depicts the HTSD when 10 heliostats directed sunlight on the solar receiver. Figure 7a shows the sensor distribution on the surface prototype (thermocouples and FsFBGs), as well as the contour points taken into account for boundary conditions at the two-dimensional interpolation algorithm (ambient temperature at these points is 25 °C). Figure 7b shows the HTSD obtained from temperature data measured by FsFBGs, using a relative colormap to represent temperature gradients. For comparative purposes, Figure 7c shows the HTSD obtained from temperature data measured by thermocouples, using an analog relative colormap. Temperature measurements reveal a maximum temperature of 195 °C for FsFBGs versus 183 °C for thermocouples. This deviation can be explained by the different positions where the sensors were located, as well as the high temperature gradients into the receiver prototype. The comparison shows minimal deviations, which is in good accordance with the estimated temperature uncertainty.

The second trial targeted the determination of HTSD with a quasi-uniform irradiance distribution and under forced air-cooling. Results for this trial are represented in Figure 8. So, Figure 8a depicts the FsFBGs temperature measurements as a function of time with 10 heliostats (from 11:00:21 to 11:10:12), 20 heliostats (from 11:17:16 to 11:29:00), 30 heliostats (from 11:34:25 to 12:03:26); and 37 heliostats (from 12:14:02 to 12:20:47). From 12:43:29 to 13:00:39, 10, 20, 30, and 37 heliostats are progressively used again. From temperature data, HTSDs are computed and represented in Figure 8b–e for times 11:10:12, 11:48:38, 12:20:47, and 13:00:39, respectively. HTSD are represented in relative and absolute terms, that means scaling the color bar from minimum to maximum at the specific time—upper figure in Figure 8b–e; and from 0 to 600 °C (lower figure in Figure 8b–e). As can be observed, the heating process by focusing heliostats leads to a quasi-uniform temperature distribution at the receiver; Figure 8b shows temperatures from 41 °C at FsFBG#4 to 166 °C at FsFBG#7, whilst Figure 8e presents temperatures from a minimum of 239 °C at FsFBG#10 to 424 °C at FsFBG#11. In Figure 8a, it is worth mentioning the heating interruption results and the ripples in temperature measurements between 11:34:00 and 12:14:03, which were caused by cloud passages on the solar field. Finally, in Figure 8a, it can be observed how FsFBG#1 only measures temperatures greater than 50.5 °C due to the limited spectral range of the optical interrogator starting at 1500 nm. A detailed analysis of the FsFBG#1 calibration function in Figure 3 corroborates this fact.

The third trial concerned the HTSD acquirement, using a single aiming point strategy (heliostats redirected sunlight to a single point on the receiver surface). Results for this trial are represented in Figure 9, where the inset of Figure 9a shows a picture of the prototype with the concentrated solar spot at its center. Figure 9a also depicts the FsFBG temperature measurements as a function of time with 6 heliostats (from 15:36:00 to 15:41:17), 8 heliostats (from 15:40:20 to 15:44:17), 12 heliostats (from 15:46:33 to 15:49:13), and 14 heliostats (from 15:50:29 to 15:55:38). From 15:59:50 to 16:03:36, 14 heliostats are simultaneously focused on the prototype. From these temperature data, HTSDs are computed and represented in Figure 9b–e for times 15:41:17, 15:49:13, 15:55:38, and 16:03:36, respectively. Figure 9 keeps the same configuration as Figure 8. In this case, at each specific time, the interval of measured temperatures is higher since the solar irradiance is concentrated and denser at the center of the prototype. Figure 9c shows that temperatures are comprised between 44 °C at FsFBG#4 and 394 °C at FsFBG#2, mainly in the zone delimited by FsFBG#1, #2, and #8. However, Figure 9e presents temperatures from 57 °C at FsFBG#10 and 566 °C at FsFBG#2, mainly located in the area delimited by FsFBG#1, #2, and #12. The high irradiances allow reaching a maximum temperature of 566 °C, similar to the optimal operation temperature in a commercial MS CRS–CSP plant.

## 4. Discussions

From the above experiments, it can be stated that multipoint FsFBG technology allows a reliable high-temperature measurement in order to obtain the temperature distribution in CRS–CSP plants. Besides, as it can be observed in Figure 5a,b, it provides a lighter and more compact solution when compared to the conventional approach based on conventional sensors, e.g., thermocouples, with a considerable cabling reduction. However, a series of critical aspects, as well as some points for improvement, have to be taken into consideration for future implementations.

With regard to the FsFBG-based optical sensor design, four points were revealed as being critical in the wavelength planning. Firstly, femtosecond laser-inscribed FBGs have shown a high tolerance in the reference wavelength peaks with respect to the specified design. As can be noticed in Table 1, absolute errors are higher than 1 nm in the fabrication process of individual FsFBGs, e.g., wavelength peak specification for FsFBG#3 was 1516.000 nm versus 1514.415 nm as wavelength reference for the fabricated FsFBG#3. Although FsFBGs present a high reflectivity (as can be observed in the reflected spectrum of Figure 2), FsFBGs do not exhibit precise phase-shift information in the fabrication process, which translates into elevated tolerances for the reference wavelength with respect wavelength specifications. Second, the spectral interval of 7.5 nm considered for wavelength shift as a function of temperature is insufficient to measure temperatures in the range from 0 to 700 °C, as shown in Figure 3. Despite considering a measurement spectral interval for FsFBG#6 from 1522.001 to 1529.600 nm (7.599 nm), this spectral interval only allows the temperature measurements between 0 and 576 °C. An increment for the spectral interval of 2 nm is necessary to measure high temperatures up to 700 °C (as shown in Figure 3 by means of the calibration functions). Spectral intervals of 10 nm for each FsFBG sensor would be desirable. Third, initial ambient temperature measurements showed deviations around 20 °C. At the initial times of Figure 8a, we can observe ambient temperature measurements between 16.18 °C measured by FsFBG#6 and 35.59 °C measured by FsFBG#12. Temperatures are obtained from wavelength shifts by the calibration functions (shown in Figure 3) so that these ones have changed along the time (from date of fabrication to date of installation). It is important to note that FsFBG sensors are not suffering stress from bending since the locations for FsFBG sensors are arranged in a straight line. In the performed experiments, any bending loss was observed in the reflection spectrum due to the temperature effect. Fourth, the optical interrogator presents a limited spectral range for operation, which translates into a reduced number of FsFBGs by channel. It would be desirable to use an optical interrogator with a wider operation spectral range (e.g., from 1460 to 1620 nm) and a greater number of channels (e.g., 2, 4, or 8 channels) with the objective to increase the number of FsFBG sensors.

With regard to the application of monitoring the HTSD at the central receiver of a CSP tower system, the set of 12 FsFBGs to obtain the HTSD has proved to be insufficient to provide a suitable resolution. In this work, each FsFBG covers 256 cm^2^ in area. This area is too large to provide an appropriate resolution with respect to the temperature gradients resulting from the heating process. In the prototype receiver, only 12 FsFBG-sensors-based measurement points were implemented in performed tests, since the optical interrogator has only one available channel. It is known that, the more measurement points that are on the prototype panel, the greater the resolution will be in the temperature map. As shown in Figure 7, there is a considerable difference between temperature maps obtained from 12 and 24 measurement points, which corresponds to FsFBGs and thermocouples, respectively. Thus, if the application needs a higher resolution, and so, a higher number of FsFBG sensors with 10 nm each as minimum measuring spectral interval, an optical interrogator with a wider spectral range and a greater number of channels will be required.

## 5. Conclusions

In conclusion, a fiber-optic sensor system was proposed for monitoring the temperature surface distribution in central receivers of CSP plants. The concept was successfully tested in a prototype specially designed for this purpose. The sensor design is based on a set of 12 FsFBGs for temperature measurements up to 700 °C. FsFBGs were inscribed in single-mode fiber and packaged with a high-temperature stainless-steel tube (stable up to 1000 °C). A central receiver prototype was developed “ad hoc” for the experiments. The prototype was instrumented with the set of 12 FsFBGs and installed at the central tower of a CSP research facility. When the central receiver prototype was exposed to different solar radiation fluxes from heliostats, characteristic temperature maps were observed. Results demonstrated the viability and reliability of the measuring technique, with temperature measurements up to 566 °C. The analysis of the design and performance of FsFBGs for high-temperature measurements showed the high tolerances in the wavelength peaks when fabricated, a minimum spectral interval of 10 nm for wavelength shifts to measure temperatures in the range of 0 to 700 °C, and the importance of calibration functions in the design and measurement processes. Upcoming investigations and future work will focus on the following: (1) analyzing the performance of regenerated FBGs as sensor element (instead of FsFBGs) to measure higher temperatures at the central receiver prototype; (2) increasing the number of sensor elements (measurement points) to improve the spatial resolution; and (3) analyzing FsFBG and RFBG calibration functions prior to the wavelength planning for multiplexing the fiber-optic sensor systems. High-temperature monitoring with FsFBG is a promising technology in many industrial areas, such us aerospace vehicles, nuclear plants, oil and gas explorations, and advanced robotics in harsh environments.

## Figures and Tables

**Figure 1 sensors-21-03762-f001:**
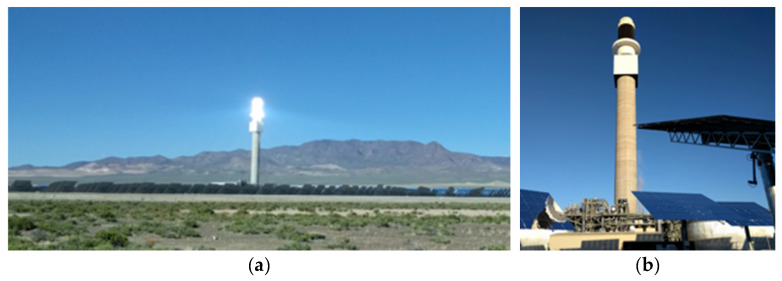
Crescent Dunes Solar Power Plant: (**a**) under operating conditions (notice the concentrated solar rays in the bright spot, where the receiver is located) and (**b**) under maintenance stop (notice the black color of the receiver, so it maximizes the optical absorptivity).

**Figure 2 sensors-21-03762-f002:**
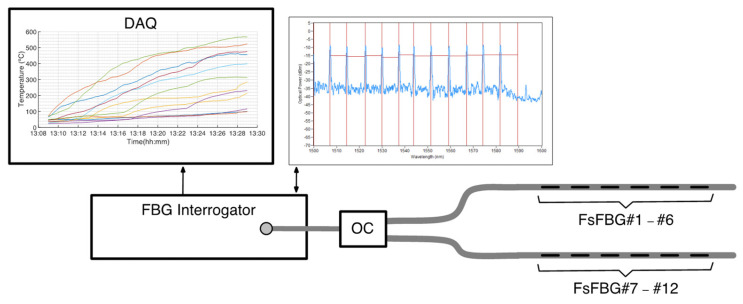
Illustrative diagram for the FsFBG-based sensor system: (OC) 3-dB Optical Coupler and (DAQ) Digital Acquisition System.

**Figure 3 sensors-21-03762-f003:**
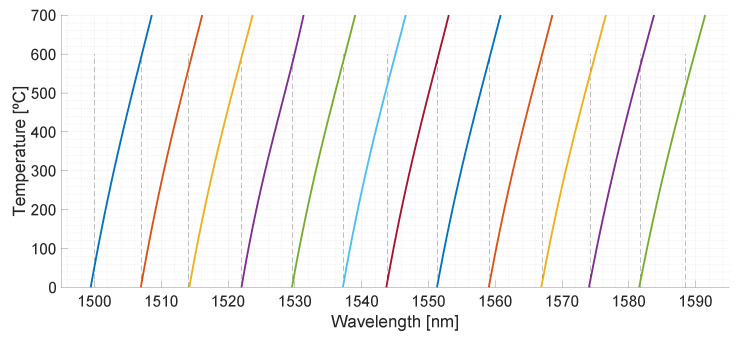
Calibration functions: peak wavelengths versus temperature for the set of the 12 FsFBGs (dashed lines delimit the spectral intervals specified in the FBG optical interrogator for the temperature measuring algorithm).

**Figure 4 sensors-21-03762-f004:**
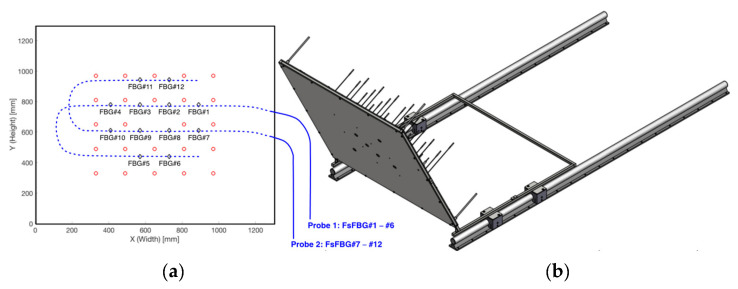
Design of the central receiver prototype: (**a**) dimensions of the prototype, layout of the two fiber-optic probes—blue lines, locations for the FsFBG#i sensors—black diamonds, and complementary thermocouples—red circles; (**b**) CAD representation for the instrumented receiver prototype.

**Figure 5 sensors-21-03762-f005:**
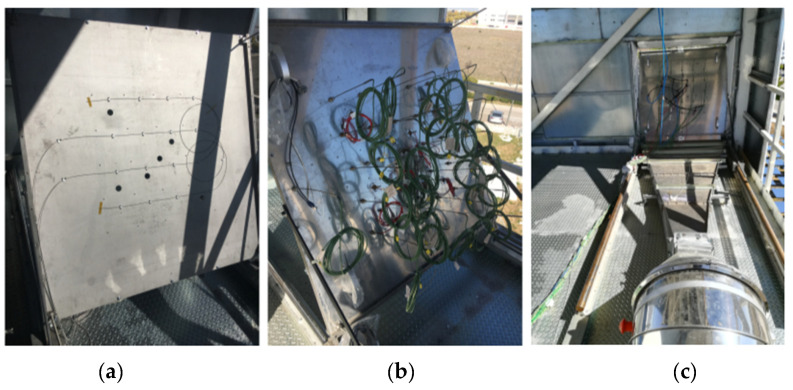
Sensor instrumentation of the central receiver prototype: (**a**) FsFBG assembly in the back of the front plate; (**b**) thermocouples assembly in the back of the rear plate; and (**c**) final arrangement for the central receiver prototype with the forced air-cooling system.

**Figure 6 sensors-21-03762-f006:**
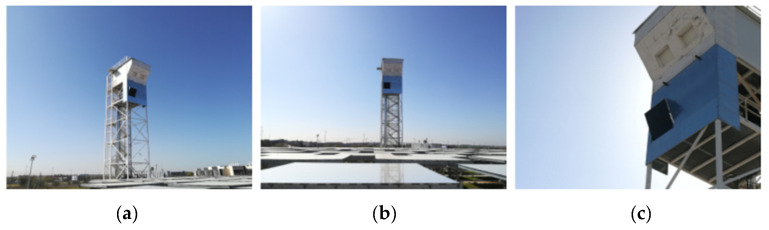
Installation of the solar receiver prototype in the VHCST facility: (**a**) lateral view of the solar tower with the central receiver prototype; (**b**) relative position of heliostat field with respect to the solar tower; and (**c**) detail of the thermal shielding and inclination of the receiver.

**Figure 7 sensors-21-03762-f007:**
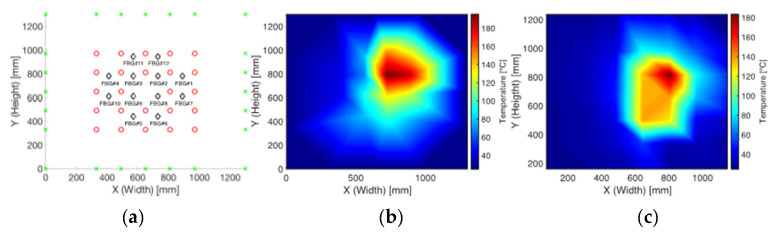
Comparison between HTSD based on FsFBGs measurements versus thermocouples measurements: (**a**) FsFBGs (black diamonds) and thermocouples (red circles) distribution on the central receiver surface plus the contour points (green cross); (**b**) HTSD from FsFBGs data; and (**c**) HTSD from thermocouples data.

**Figure 8 sensors-21-03762-f008:**
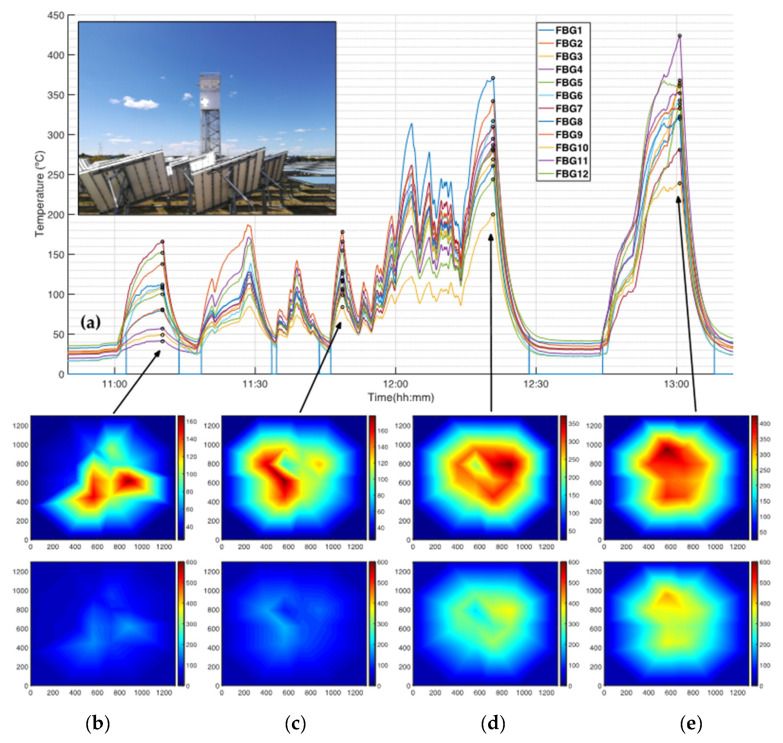
FsFBGs temperature measurements and corresponding HTSDs at the central receiver prototype for quasi-uniform flux distributions at the prototype receiver: (**a**) FsFBGs temperature measurements as a function of time (the inset represent the concentrated solar radiation at the receiver prototype to obtain quasi-uniform flux distributions); (**b**) HTSD from FsFBGs temperature data at 11:10:12—up, using a relative colormap, and down using an absolute colormap; (**c**) HTSD from FsFBGs temperature data at 11:48:38—up and down, using a relative and absolute colormap, respectively; (**d**) HTSD from FsFBGs temperature data at 12:20:47—up and down, using a relative and absolute colormap, respectively; (**e**) HTSD from FsFBGs temperature data at 13:00:39—up and down, using a relative and absolute colormap.

**Figure 9 sensors-21-03762-f009:**
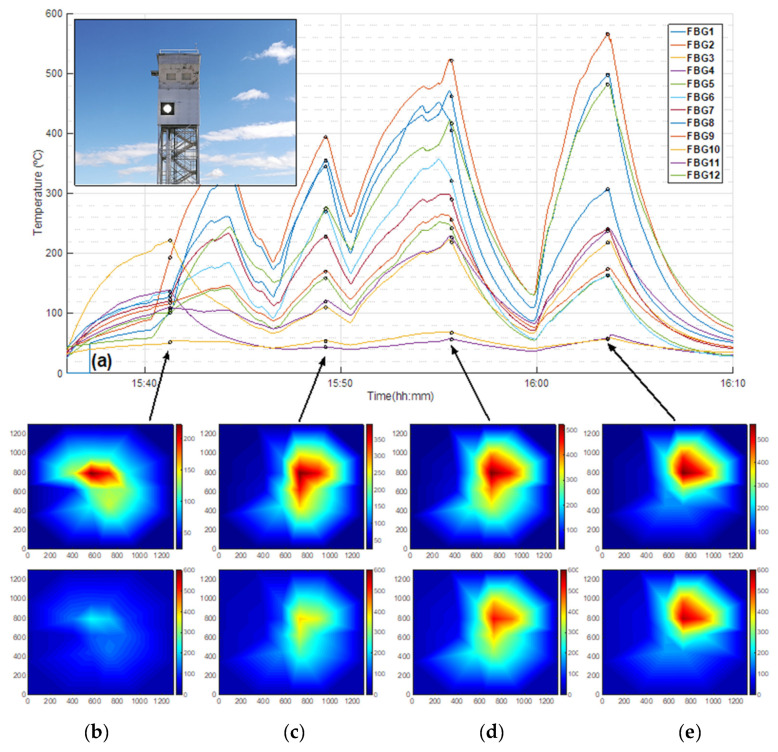
FsFBGs temperature measurements and corresponding HTSDs for concentrated high-flux distributions at the central receiver prototype: (**a**) FsFBGs temperature measurements as a function of time (the inset represent the concentrated solar radiation at the receiver prototype to obtain concentrated flux distributions); (**b**) HTSD from FsFBGs temperature data at 15:41:17—up, using a relative colormap, and down using an absolute colormap; (**c**) HTSD from FsFBGs temperature data at 15:49:13—up and down, using a relative and absolute colormap, respectively; (**d**) HTSD from FsFBGs temperature data at 15:55:38—up and down, using a relative and absolute colormap, respectively; (**e**) HTSD from FsFBGs temperature data at 16:03:36—up and down, using a relative and absolute colormap.

**Table 1 sensors-21-03762-t001:** Wavelength planning for the FsFBG-based fiber-optic sensor system.

	Wavelength Peak (nm)		Sensitivity ^2^ (pm/°C)	Spectral Interval (nm)	
	Specification	λ_ref_ ^1^		From	To
FsFBG#1	1501.0	1549.655	12.91	1500.000	1507.0
FsFBG#2	1508.5	1507.154	12.97	1507.001	1514.0
FsFBG#3	1516.0	1514.415	13.33	1514.001	1522.0
FsFBG#4	1523.5	1522.220	13.35	1522.001	1529.6
FsFBG#5	1531.0	1529.780	13.38	1529.601	1537.3
FsFBG#6	1538.5	1537.409	13.26	1537.301	1543.8
FsFBG#7	1546.0	1543.998	13.28	1543.801	1551.4
FsFBG#8	1553.5	1551.590	13.51	1551.401	1559.2
FsFBG#9	1561.0	1559.301	13.51	1559.201	1567.0
FsFBG#10	1568.5	1567.160	13.66	1567.001	1574.2
FsFBG#11	1576.0	1574.288	13.71	1574.201	1581.7
FsFBG#12	1583.5	1581.811	13.91	1581.701	1588.5

^1^ Wavelength peaks for fabricated FsFBGs measured at 22 °C. ^2^ Average sensitivity for fabricated FsFBGs between 0 and 600 °C.

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
