# Peer review of "High-Temperature Monitoring in Central Receiver Concentrating Solar Power Plants with Femtosecond-Laser Inscribed FBG"

_sensors, 2021, doi:10.3390/s21113762_

Round 1

Reviewer 1 Report

This paper investigates high temperature monitoring in central receiver concentrating solar power plants by using the FBG made by femtosecond-laser . On the review, I think this paper should make minor revision after accept. The comments are as following.

  1. The author should give the manufacture processing for the FBG by femtosecond-laser.
  2. The layout of the FBG in the solar power plants is not very clear and should be offered more detailed.
  3. The Fig.7a presents some noise signal when the FBG  measure the temperature signal. The author should give more scientific explanation.
  4. The materials of FBG will suffer the thermal effect during the measuring process. How could the author to avoid except for using the cooling system.

Reviewer 2 Report

With reference to the Manuscript No.- Sensors-1225164, Paper Titled - High-Temperature Monitoring in Central Receiver Concentrating Solar Power Plants with Femtosecond-laser inscribed FBG, I would like to propose my few comments to this paper. 

Comments:

  1. Try to provide more reference under the Introduction section for CSP, if possible.
  2. Try to mention temperature sensitivity of each FBGs or FBGs as a device in a package of 6FBGs altogether, if possible.
  3. Line 154-155: What is the reason for choosing INCOLOY 625 and PYROMARK 2500 for plates and as a primer respectively.
  4. Line 160: What does the Transfer fluid performs? Briefly mention if possible.
  5. Line 197: Briefly explain the reason for using a ceramic-plates for protective thermal shielding.
  6. Line 228: Try to clearly explain the under forced air-cooling system, if possible.
  7. Line 41: Replace “trough” to “Through”.
  8. Line 90: Replace “Gratins” to “Gratings”.
  9. Line 105: “Design” word is repeated.
  10. Line 301: Did you perform or check any bending loss for these FBGs due to temperature effect.
  11. Try to improve the conclusion section mentioning the probable application domain of your research.

Round 2

Reviewer 1 Report

All comment have been modified. The paper could be accepted as this revised form.